# Microglia and Brain Macrophages as Drivers of Glioma Progression

**DOI:** 10.3390/ijms232415612

**Published:** 2022-12-09

**Authors:** Yuqi Zheng, Manuel B. Graeber

**Affiliations:** Ken Parker Brain Tumour Research Laboratories, Brain and Mind Centre, Faculty of Medicine and Health, University of Sydney, Camperdown, NSW 2050, Australia

**Keywords:** epigenetics, exosomes, “Janus” genes, “Janus” pathways, microglia, monocyte, macrophage populations, microRNA

## Abstract

Evidence is accumulating that the tumour microenvironment (TME) has a key role in the progression of gliomas. Non-neoplastic cells in addition to the tumour cells are therefore finding increasing attention. Microglia and other glioma-associated macrophages are at the centre of this interest especially in the context of therapeutic considerations. New ideas have emerged regarding the role of microglia and, more recently, blood-derived brain macrophages in glioblastoma (GBM) progression. We are now beginning to understand the mechanisms that allow malignant glioma cells to weaken microglia and brain macrophage defence mechanisms. Surface molecules and cytokines have a prominent role in microglia/macrophage-glioma cell interactions, and we discuss them in detail. The involvement of exosomes and microRNAs forms another focus of this review. In addition, certain microglia and glioma cell pathways deserve special attention. These “synergistic” (we suggest calling them “Janus”) pathways are active in both glioma cells and microglia/macrophages where they act in concert supporting malignant glioma progression. Examples include CCN4 (WISP1)/Integrin α6β1/Akt and CHI3L1/PI3K/Akt/mTOR. They represent attractive therapeutic targets.

## 1. Introduction

This article focuses on microglia and brain macrophages in glioblastoma (GBM), but many findings are expected to be relevant for lower grade gliomas as well. Microglia and recently blood/bone marrow-derived macrophages (BMDM) constitute the largest population of glioma infiltrating cells (e.g., [1]) and it is now accepted that their increased presence correlates with glioma progression and reduced patient survival [2]. Interestingly, there are significant differences between the two cell populations and also in between microglia cells within the tumour microenvironment (TME) [3,4,5,6]. Microglia and brain macrophages are differentially distributed in GBM tissue and much has been learned about their recruitment and functional qualities over the last decade [4,5].

## 2. Heterogeneity of Cells in TME

It has been demonstrated that microglia and BMDM display distinct phenotypic signatures and localisations within GBM indicating complexity and diversity of the myeloid compartment in malignant glioma [7,8].

### 2.1. Macrophage Scavenger Receptor 1 (MSR1/CD204)

It is becoming increasingly clear that endothelial cells are a key component in organising the perivascular niche and sustaining the survival and stemness of glioma stem cells GSCs [9]. Recent accumulating evidence indicated that GSCs predominantly reside in a perivascular niche [10,11] and usually surround by tumour-associated macrophages and microglia [12]. A previous study by Yi et al. has further reported that GSCs are better at recruiting microglia than glioma cells as GSCs expressed more Chemokine (C-C motif) ligand 2 (CCL2), Chemokine (C-C motif) ligand 5 (CCL5), Chemokine (C-C motif) ligand 7 (CCL7), Vascular endothelial growth factor A (VEGF-A) and neurotensin (NTS) than glioma cells that infiltrate the brain tissue [12]. In addition, tumour associated macrophages and microglia (TAMs) maintain the phenotype of glioma-stem cells by releasing TGF-β1, which in turn promotes glioma growth and invasion [13]. MSR1 (CD204) is a class A macrophage scavenger receptor and pattern recognition molecule associated with a pro-tumourigenic phenotype of brain macrophages [14]. High levels of MSR1+ (CD204) are associated with increased expression of immune checkpoint markers PD-L1 and TIM3 which raise the possibility that MSR1+ may contribute to T cell exhaustion [15,16]. A recent study employing single-cell RNA sequencing (scRNA-seq) reported that while BMDM aggregate perivascularly and within necrotic foci, microglia expression signatures are enriched within the glioma infiltration front (“tumour edge”). Furthermore, the increased presence of blood-derived tumour-associated macrophages correlated with shorter survival times [7]. Sørensen and Zhang et al. have shown that the number of MSR1+ glioma-associated microglia/macrophages increases with malignancy grade [14,17]. A later study performed mRNA transcriptome profiling followed by pathway and connectivity network analysis (STRING, https://www.string-db.org/; accessed on 1 November 2022) revealed that the accumulation of MSR1+ microglia/macrophages in glioblastoma correlates with an interleukin-6-enriched profile and poor survival [18] (Figure 1, Molecular event 7). Sørensen and Kristensen [18] further demonstrated that MSR1+ microglia/BMDM accumulate perivascularly and around necrotic areas and that they often co-reside with stem-like GBM cells expressing the marker, podoplanin (PDPN) [18]. Thus, MSR1+ microglia/BDMDs may support stem-like GBM cells and the progression of GBM in perivascular and necrotic niches.

### 2.2. Purinergic Receptor P2Y12 (P2RY12)

Results from recent flow cytometry experiments indicate that P2RY12 and CD49 may be used to distinguish microglia from BMDM in GBM [7,8]. In line with this, a single-cell image analysis study by Woolf et al. [47] reported that P2RY12 and TMEM119 label microglia in GBM and the authors further proposed that the markers can be used to discriminate microglia from BMDM. Of note, patients with high P2RY12 expression survived longer. Moreover, a higher microglia to BMDM ratio in GBM conferred a survival advantage that was independent of O-6-methylguanine-DNA methyltransferase (MGMT) methylation status [47]. Interestingly, activation of P2Y12 receptors causes extension of microglial cell processes [48,49]. It is noteworthy that phagocytic GBM-associated microglia and macrophages have also been observed in the non-necrotic parts of (pseudo)palisading GBM necrosis [48,49]. Importantly, microglia/macrophages in the GBM resection zone have been suggested to function as part of a glioma stem cell niche at the tumour border [50,51] which is important for tumour recurrence. It is suggested that they express a distinct gene signature [52]. The supportive influence of microglial cells on glioma growth is now established beyond doubt and can even be reproduced in a xenograft model [52]. Interestingly, microglia seem to exert a sex-specific influence in the TME [53] which may help to explain why GBM is more common and aggressive in male patients. In fact, the hijacking of sexual immune privilege by GBM has been identified as an immune evasion strategy of the glioma [53,54,55].

## 3. Mechanisms Underlying the Recruitment of Microglia and BMDM into Glioma Tissue

We have previously reviewed the role of CCL2, Hepatocyte growth factor/scatter factor (HGF/SF), VEGF and Macrophage colony-stimulating factor (M-CSF) [56]. Much novel information has been discovered in this field of research in recent years. For instance, Chang et al. have found that macrophages and microglia associated with glioma produce CCL2 which is considered critical for recruiting regulatory T cells and myeloid-derived suppressor cells into the tumour microenvironment [57]. Moreover, the authors reported that CD163-immunoreactive infiltrating macrophages are a major source of CCL2 [57]. CCL2 has also been implicated in macrophage recruitment into GBM that is stimulated by EGFR and EGFRvIII [58].

### 3.1. Secreted Phosphoprotein 1 (Spp1)

Secreted phosphoprotein 1 (Spp1), also known as osteopontin (OPN), is a glycophosphoprotein expressed by various cell types, including macrophages, T-cells, osteoblasts, epithelial cells and tumour cells. Moreover, expression of Spp1 can regulate cell-matrix interactions by binding to CD44 and integrin receptors thus mediating cell adhesion, chemotaxis, angiogenesis and resistance to apoptosis [59]. Notably, Spp1 expression which like CD68 [60] is especially high in mesenchymal GBM, correlates with both tumour grade and the extent of macrophage infiltration. GBM and glioblastoma stem cells (GSCs) use Spp1 to recruit macrophages into the TME [19]. Integrin αvβ5 (ITGαvβ5), a key receptor for Spp1, is highly expressed on GBM-infiltrating macrophages [19] (Figure 1, Molecular event 1). This is in line with previous findings by Ellert-Miklaszewska et al. that Spp1 and lactadherin enable glioma cells to gain an advantage through M2 reprogramming of tumour-infiltrating brain macrophages [61]. Therefore, Spp1 is likely to play a key role in the progression of GBM.

### 3.2. Periostin (POSTN)

POSTN is a disulfide-linked cell adhesion protein which belongs to the fasciclin (Fas) family [62]. Recent studies have revealed that POSTN contributes to malignant tumour progression by supporting metastatic colonisation of breast cancer stem cells through upregulation of Wnt signalling [63]. A study by Zhou et al. [20] has suggested that BMDM are the main source of macrophages in GBM and that their recruitment from the bloodstream is stimulated by GBM-secreted periostin (POSTN) that interacts with integrin αvβ3 (ITGαvβ3) on BMDMs (Figure 1, Molecular event 2). As expected, *POSTN* knockout mice show significantly extended survival times. It is noteworthy that POSTN is preferentially expressed by putative GSCs expressing SOX2 and OLIG2, respectively [20]. This is in keeping with the results of Guo et al. [64] who reported that hypoxia promotes glioma-associated macrophage infiltration via POSTN.

### 3.3. Nuclear and Secreted IL-33

IL-33, a member of the IL-1 cytokine family, is secreted as an alarmin by damaged or necrotic cells [65]. It is now clear that IL-33 plays a pro-tumorigenic role in various cancers including glioma [65]. De Boeck and colleagues [34] recently suggested that both the nuclear and secreted form of IL-33 are present within tumour cells in ~50% of human glioma specimens and GBM murine models. In addition, IL-33 was previously shown to associate with chromatin and be involved in the regulation of gene transcription [66]. Using multiplex cytokine/chemokine analysis, these authors further demonstrated that nuclear IL-33 facilitates tumour growth by triggering glioma-mediated expression of inflammatory cytokines (LIF, IL-6, IL-8, IL-1RN, IL-1β and secreted IL-33 [34]. Strikingly, a subset of microglia in the IL-33+ xenografts also expressed a significant amount of IL-33 and showed enrichment in pro-inflammatory cytokines, Spp1 (osteopontin), and the lipid metabolism gene, *Apoe*. These microglial cells further displayed a notable upregulation of the monocyte chemoattractant genes (*CCL2*, *CCL3*, and *CCL12*). Activation of monocyte chemoattractant genes fuels further recruitment of immune cells to the glioma microenvironment [34] (Figure 2, Molecular event 16). Conversely, loss of nuclear IL-33 resulted in significantly smaller IL-33-associated tumour burden and increased overall survival. Furthermore, elevated levels of both nuclear and soluble IL-33 were associated with enhanced activation of AIF1 (Iba1)+ resident microglia and recruitment of CD163+ BMDM [34]. In addition, neutralising IL-33 effects by-means of anti-IL-33 and anti-CCL2 (which is up-regulated by recombinant IL-33) antibodies significantly reduced the recruitment of microglia and BMDM [34].

### 3.4. Programmed Cell Death Protein 10 (PDCD10)/CXC Motif Chemokine Ligand 2 (CXCL2)/CXCR2 Signalling

PDCD10 is an evolutionarily conserved protein expressed by neurons, astrocytes, endothelial and cancer cells. Zhang et al. [37] recently reported that overexpression of PDCD10 by GBM cells promotes tumour progression via recruitment of microglia and BMDM. Furthermore, PDCD10 up-regulation is followed by an increase in CXC motif chemokine ligand 2 (CXCL2) and resulting activation of CXCR2 in microglia (Figure 3, Molecular event 20). In sum, CXCL2-CXCR2 signalling stimulated by PDCD10 appears to be a key mechanism in the crosstalk between GBM cells and microglia/macrophages that promotes tumour progression.

### 3.5. Slit Guidance Ligand 2 (SLIT2)-Roundabout 1 and 2 (ROBO1/2)

SLITs are evolutionarily conserved polypeptides that bind to cells expressing ROBO receptors. SLIT-ROBO binding activates the recruitment of adaptor protein to the cytoplasmic domain of ROBO receptors which in turn regulates cell motility by modulating the actin and microtubule cytoskeleton [84]. Recently, SLIT2 has been found to promote microglia/macrophage chemotaxis (via ROBO1/2 induced PI3Kγ activation) and polarization, and its expression to increase with malignant progression and correlate with poor survival and immunosuppression [21] (Figure 1, Molecular event 3). It is worth noting that *SLIT2* knockdown in tumour cells inhibits mouse macrophage invasion [21].

### 3.6. Lethal-7 microRNAs (Let-7 miRNA)

The family of let-7 microRNAs, which share an evolutionarily conserved sequence, are highly expressed in the brain [85]. Recent studies have indicated that let-7 microRNAs are involved in cancer initiation and brain tumour progression [85]. Moreover, it is known that Toll-like receptors are pattern recognition receptors found on microglia that detect pathogen- and host-derived factors such as miRNAs [85]. Let-7 miRNA, specifically ones carrying the core sequence motif UUGU, can activate microglial and BMDM Toll-like receptor (TLR) 7 and induce TNF-α production which might lead to suppression of glioma growth [68]. It is worth noting that selective groups of let-7 miRNAs regulate the expression of antigen-presenting molecules in the CNS. These includes let-7b and let-7e miRNAs which stimulate up-regulation of MHC I and ICAM1 (CD54) through TLR7 signalling [68]. Of note, MHC I and ICAM1 are critically important for the communication between innate and adaptive immune cells and activation of T cell-mediated cytotoxic responses [68]. Interestingly, let-7 miRNA oligoribonucleotides that lack the GU-rich core motif may act as a chemoattractant for microglial cells in glioma [68] (Figure 2, Molecular event 12). Furthermore, the let-7 miRNAs induced TLR-7 activation is observed in both neonatal and adult microglia/BMDMs [68]. Thus, it is hypothesised that let-7 miRNAs can potentially shift microglia towards an “anti-tumour” phenotype and enhance the efficacy of immunotherapies.

## 4. Glioblastoma Weakens Microglia/Macrophage Defence Mechanisms

### 4.1. Molecules Involved in the Differentiation of Monocytes

#### 4.1.1. CD14 and MicroRNA-146 (miRNA-146)

Numerous studies have demonstrated that GBMs are infiltrated by immune cells, and by microglia and monocyte-derived macrophages in particular [86]. It has been shown that an increased number of microglia and brain macrophages is associated with a higher WHO grade in gliomas [87]. The WHO classification is essentially a malignancy scale that helps clinicians predict a patient’s disease course (prognostication). Tumors are graded benign (grade 1) or malignant, and there are different grades of malignancy expressed as “grade 2–4” with 4 being worst (shortest expected survival time) [88]. In order to avoid confusion, a grade 2 glioma will progress and cannot be considered benign although it is not fully malignant yet. Gabrusiewicz et al. recently reported that the number of CD14+ monocytes is increased in the blood of GBM patients [35]. Using whole-genome expression profiling, the authors also observed that GBM-associated myeloid cells do not exist in distinct polarized M1 and M2 states. Moreover, gene set enrichment analysis revealed MYC and E2F transcriptional regulation in CD14+ cells. Interestingly, Gabrusiewicz et al. also pointed out that miRNA-146 may play a role in pro-inflammatory macrophages (Figure 2, Molecular event 10); its expression was significantly suppressed in GBM-infiltrating CD14+ cells. Strikingly, a previouly mentioned glioma-derived molecule, Spp1 was also highly expressed in GBM-infiltrating CD14+ BMDM [35].

#### 4.1.2. Macrophage Migration Inhibitory Factor (MIF)-CD74

MIF is a molecule that is highly conserved across species suggesting it has a role in fundamental biological processes [89]. Accumulating evidence suggests that MIF is expressed by immune cells in various cancers including breast and lung cancer [89]. Myeloid-derived suppressor cells (MDSCs) are a heterogeneous group of bone marrow-derived progenitor cells that consist of monocytic (M-MDSC) and granulocytic (G-MDSC) subsets which exhibit potent immunosuppressive activity. They interfere with the cytotoxic functions of natural killer (NK) cells and T lymphocytes in tumours including GBM [90,91]. A recent experimental study by Alban and colleagues [92] has found that the monocytic subset of myeloid-derived suppressor cells (M-MDSCs) expresses high levels of CD74 in the presence of glioma-derived macrophage migration inhibitory factor (MIF) and glioma cells (Figure 1, Molecular event 5). Using a syngeneic murine model, the authors further described that disruption of the MIF-CD74 pathway using Ibudilast minimises downstream activation of CCL2 (MCP-1). CCL2 was previously shown to have a critical role in driving recruitment of monocytes and expansion of MDSCs [93,94]. Of note, MIF is also capable of mediating signalling via non-cognate receptors such as CXCR2, CXCR4, and CXCR7 [92]. Importantly, activation of microglial CD74 weakens the microglial defense against glioma cells [23]. Furthermore, MIF expression is significantly increased in malignant glioma and interferon (IFN)-γ secretion by microglia is inhibited by MIF-CD74 signalling [23].

### 4.2. GBM-Induced Impediment of Microglia/BMDM Phagocytic Activity

#### 4.2.1. P-Selectin (SELP)-P-Selectin Glycoprotein Ligand-1 (PSGL1)

SELP is a well-known adhesion molecule involved in leukocyte rolling and recruitment [95]. It is now evident that SELP and its ligand PSGL-1 are involved in the metastatic spread of melanoma and colon cancer [96]. Recently, a mechanism has been proposed by which SELP-PSGL1 mediate GBM progression influencing microglia/macrophage phenotype [22]. The authors found that recombinant SELP reduced the phagocytic activity of microglia/BMDM, decreased their expression of inducible nitric oxide synthase (iNOS) and release of nitric oxide (NO) while increasing expression of IL-10 and TGF-β. It is worth noting that following exposure to soluble SELP (sSELP), a positive feedback loop causes overexpression of SELP and PSGL-1 by GBM and microglia cells [22] (Figure 1, Molecular event 4). On the flip side, expression of actin nucleation promoting factor wasla by microglial cells was found to revive microglial phagocytotic activity and slow down GBM progression in zebrafish [97].

#### 4.2.2. CD47-SIRPα Anti-Phagocytic Axis

A recent experimental study by Hutter and co-workers has indicated that tumour-associated microglia are capable of tumour cell phagocytosis in vivo if the immune evasion of tumour cells is blocked by a humanized anti-CD47 monoclonal antibody [29]. Notably, Li et al. have shown that CD47 is expressed by human and mouse glioma cell lines and that positive cells have many characteristics of cancer stem cells [98]. The view is also supported by Hu et al. [99] who report that overexpression of the *LRIG2* (Leucine Rich Repeats And Immunoglobulin Like Domains 2) gene in GBM cells induces upregulation of CD47 and activation of the CD47-SIRPα anti-phagocytic axis [30]. Further experiments revealed that soluble LRIG (sLRIG) induces recruitment of BMDM that exhibit an immunosuppressive phenotype and also express high levels of CD47 receptor, SIRPα [99]. As expected, knockdown of *LRIG2*/sLRIG2 in GL261 (murine GBM) cells interferes with the activation of the CD47–SIRPα anti-phagocytic axis and enhances BMDM-mediated phagocytosis of GBM cells and suppresses GBM progression [99]. This is in keeping with the results of a xenograft study by Gholamin et al. showing ubiquitous expression of CD47 in paediatric GBM and diffuse midline glioma [30]. Blockage of the anti-phagocytic CD47-SIRPα axis using an anti-CD47 antibody, Hu5F9-G4, strongly induced BMDM-mediated phagocytosis of glioma cells, and mice treated with Hu5F9-G4 demonstrated significant longer survival [30] (Figure 1, Molecular event 9). It is worth noting that the “don’t eat me” signal mediated by the CD47-SIRPα axis [100] also protects synapses from non-specific pruning during development and disease. Accordingly, CD47 deficiency in mice leads to reduced synaptic density resulting from excessive pruning by microglia [31]. Loss of microglial SIRPα has a similar effect in preclinical models of neurodegeneration [101]. Thus, the CD47-SIRPα axis may deserve special attention in the context of cognitive deficits of brain tumour patients even though Li et al. were unable to detect damage to neurons and astrocytes in a treatment model [98].

### 4.3. GBM-Induced Immune Tolerance Involving Microglia/BMDM

The interactions between tumour-associated microglia/brain macrophages and T cells may lead to T cell malfunction and diminished T-cell mediated anti-tumour responses [102]. By analysing RNAs found in extracellular vesicles (EVs) derived from microglia that had interacted with GBM, Maas et al. [103] recently found that GBM-interacting microglia down-regulate genes involved in the detection of tumour cells (sialic acid-binding immunoglobulin-like lectin-H (*Siglec-H*), *CD33* and *GPR34*) and tumour-derived metabolic by-products (*Gpr183*, *Adora3*, *Il6Ra*, *Cx3cr1*, *P2ry12*, *P2ry13*, *Csf1r*, and *Csf3r*) [103]. In contrast, levels of *CD274* (PD-L1) and *PD-L2* transcripts are elevated in GBM-interacting microglia, suggesting that genes involved in immunologic tolerance are up-regulated in microglia by GBM contact, resulting in indirect inhibition of anti-tumour functions of T cells [104]. Interestingly, the authors further demonstrated that GBM-interacting microglia show up-regulated expression of phagocytic receptors (*Cd93*, *Msr1*, *Cd36*, *Olr1*, *Megf10*, *Clec7a*, *Scarf1*) and extracellular matrix (ECM) degrading enzymes such as *Mmp14* [103]. A recent study supports the invasion-facilitating role of microglial cells by describing that glial cell line-derived neurotrophic factor (GDNF), a chemoattractant of microglia [105], stimulates the production of microglia-derived MMP9 and MMP14 in neonatal mice [106]. Huang et al. further indicated that GDNF induced up-regulation of microglial TLR1 and TLR2 and that the activation of TLR2 can increase expression of microglial MMP9 and MMP14 [106]. In addition to stimulating expression of ECM degrading enzymes, the activation of TLR2 also inhibited expression of MHC class II by microglial cells via loss of histone H3 acetylation at the master regulator of MHC class II molecule transcription, *Ciita* (Class II Major Histocompatibility Complex Transactivator). Accordingly, inhibited MHC II expression impedes CD4+ T cell activation and proliferation which weakens T-cell dependent anti-tumour responses [107]. Expression of the GBM-associated microglial phenotype appears to be mediated by EVs, a view that is supported by animal experiments demonstrating that intracranial injection of glioma-derived EVs in healthy mice results in similarly modified transcription [103]. Recently, Mirzai and Wong [5] have reviewed microglia-T cell communication and pointed out that the synthesis of immunosuppressive cytokines such as TGF-β and IL-10 is increased as a consequence of their interaction. In addition to the release of immunosuppressive cytokines, *Acod1* (aconitate decarboxylase 1) has been identified as a gene that is involved in the regulation and subsequent adjustment of the microglia/macrophage phenotype during GBM progression [108]. These findings are in line with the view that GBM alters gene transcription in microglia, supporting tumour invasion and migration while microglia remove necrotic debris and digested ECM in the TME. It is also worth mentioning that CXCL14 has been proposed as an important determinant of the glioma immune microenvironment where it is thought to promote activated CD8+ T cell chemotaxis which appears to prolong survival [109]. Interestingly, pleomorphic xanthoastrocytoma (PXA) shows increased CXCL14 secretion and contains a higher number of activated cytotoxic CD8+ T cells, increased expression of MHC class I and other genes associated with antigen presentation and processing as well as a higher number of AIF1 (Iba1)+ immunoreactive microglia/macrophages when compared to IDH-mutant astrocytoma [109]. Blockage of Spp1 not only reduces recruitment of macrophages but also renders GBM cells more sensitive to direct CD8+ T cell cytotoxicity [19]. Inhibition of MIF-CD74 interaction also leads to the expansion and activation of CD8+ T cells [92].

## 5. Microglial/BMDM-Derived Factors Supporting Glioma Progression

### 5.1. C-C Motif Chemokine Ligand 5 (CCL5)/Akt/Calcium (Ca^2+^)/Calmodulin-Dependent Protein Kinase II Phosphorylation (p-CaMKII) Pathway

CCL5 is an inflammatory cytokine secreted by multiple cell types, including endothelial cells, monocytes, macrophages and NK cells [110]. Moreover, CCL5 is involved in tumour growth and cell migration in various cancers including glioma [110]. Glioma cells that have been stimulated with CCL5 show increased intracellular calcium levels and elevated Akt (p-Akt) and Ca^2+^/calmodulin-dependent protein kinase II phosphorylation (p-CaMKII) in a time- and dose-dependent manner. Increased intracellular calcium levels and p-CaMKII lead to upregulated expression of calcium-dependent MMP2 in glioma cells [81] (Figure 3, Molecular event 29); MMP2 has been previously associated with GBM cell migration and invasion [111]. In addition to the invasion-promoting role of CCL5, Wu et al. also demonstrated that glioma cells exhibit a strong affinity for glioma-associated microglia/macrophages (GAMs), specifically GAMs that have been activated by Granulocyte-macrophage colony-stimulating factor (GM-CSF). The authors even use the term “homing”. This observation fits with findings showing that conditioned media derived from GM-CSF activated GAMs contain significantly higher concentrations of CCL5 [81] (Figure 3, Molecular event 29). Moreover, a more recent study has shown that expression of CD11a by microglia may play an important role in the production of glioma derived CCL5 [112]. On the flip side, small interfering RNA silencing of *CaMKII* resulted in inhibition of CCL5-mediated glioma invasion [81]. Furthermore, downregulated expression of microglial CCL5 and CCR2 in athymic mice showed impaired engraftment of *Nf1* optic low grade glioma stem cells [113]. Another study has shown that Na+/H+ exchanger isoform 1 (NHE1), a major interaction partner of calmodulin, stimulates microglial release of soluble factors leading to enhanced glioma proliferation and invasion [114] whereas blockage of NHE1 improves glioma tumour immunity by restoring mitochondrial OXPHOS (oxidative phosphorylation) function in myeloid cells [115]. Furthermore, Venkataramani et al. [116] have found that neural stimulation induces higher intracellular calcium level in GBM cells resulting in de novo formation of GBM microtubes and increased tumour invasiveness.

### 5.2. Stress Induced Phosphoprotein 1 (STIP1/STI1)

The co-chaperone STIP1 (STI1), a ligand of the cellular prion protein [117], has been demonstrated to participate in the survival and differentiation of neuronal cells [118]. STIP1 is highly expressed in glioma cells [46]. Strikingly, increased levels of STIP1 are also noted in microglia/macrophages as glioma progresses [46]. Furthermore, a significant upregulation of STIP1 expression is observed in glioma-infiltrating macrophages [46]. Therefore, *STIP1* falls into the category of “Janus” genes (please see below and Table 1). They represent promising therapeutic targets.

## 6. Isocitrate Dehydrogenase (IDH) Mutation Status Influences Glioblastoma Microglia/Macrophage Tissue Phenotype

Large-scale histological and molecular genetic studies have demonstrated that *IDH*-wildtype (*IDH*-WT) GBM is the most common and aggressive glioma subtype. In comparison, glioma patients that carry an *IDH1/2* mutation (*IDH*-mutation) show comparatively longer survival times [88]. These differences are reflected in the new (2021) WHO classification of CNS tumours by the creation of separate categories for these tumour types [88]. Importantly, the observation of co-expression of *IDH1^R132H^* and the macrophage marker CD68 in human GBM specimens by Cao et al. stimulates renewed interest in the possible existence of TAM-GBM cell hybrids [119].

### 6.1. Microglia/Macrophages in IDH-Mutant Astrocytoma (Grade 4) and IDH-Wildtype GBM

By leveraging single-cell transcriptomics, Liu et al. [77] have pointed out that the presence of a higher number of microglia/macrophages correlates with a worse prognosis in *IDH*-wildtype (*IDH*-WT) GBM. With respect to *IDH*-WT GBM, Klemm et al. [120] found that there is a difference in the ratio between microglia and BMDM between the different high grade glioma subtypes with microglia being more abundant in *IDH*-mutant gliomas [120]. In line with findings by Liu and Klemm et al., Poon and colleagues [121] have observed that there are strikingly fewer microglia and macrophages in grade 4 *IDH*-mutant astrocytoma than in *IDH*-WT GBM. Interestingly, these cells show a pro-inflammatory signature in *IDH*-mutant astrocytoma. In contrast, a macrophage anti-inflammatory phenotype (upregulation of *FCER1G* and *TYROBP* genes) was found in *IDH*-WT GBM [121]. It has been demonstrated that *FCER1G* and *TYROBP* genes play a key role in the *CSF1R* pathway [122] and are essential for the differentiation of microglia and macrophages [123]. Work using experimental animals suggests that the immunosuppressive microenvironment of *IDH1*-WT GBM can be influenced by blocking Wnt signalling between microglia and cancer cells [124].

### 6.2. SET Domain Containing 2, Histone Lysine Methyltransferase (SETD2)

A recent scRNA-seq analysis study has reported that NLR family pyrin domain containing 1 (NLRP1) inflammasome mediated IL-1β expression by microglia induces proliferation of GBM cells [77]. Microglia in mutant *SETD2* (*SETD2*-mut)/*IDH*-WT GBM exhibit pro-inflammatory and proliferative phenotypes probably through stimulation of glioma-derived TGF-β1 expression via the apolipoprotein E (ApoE)-mediated NLRP1 inflammasome [77]. Of note, TGF-β1/TGF-β receptor I (TbRI) depletion might be used to reduce the density of microglia and to suppress tumour growth [77] (Figure 3, Molecular event 25).

### 6.3. Intercellular Adhesion Molecule 1 (ICAM1), Lysosomal Associated Membrane Protein 1 (LAMP1) and Transmembrane Protein 119 (TMEM119)

A xenograft study revealed that introduction of a heterozygous *IDH1^R132H^* mutation into glioma cells has an effect on glioma-associated macrophages which are stimulated to express a more phagocytic, anti-tumour phenotype [76]. These authors further suggested that the underlying mechanism appears to involve ICAM1 [76]. Mutant *IDH1* (*IDH1*-mut) glioma cells downregulate *ICAM1* via *ICAM1* promoter methylation resulting in an increased expression of LAMP1 (CD107a), a lysosome-associated membrane protein which has a key role in the formation of phagolysosomes [125] (Figure 3, Molecular event 24). The authors also found that the microglia marker TMEM119 was decreased significantly in *IDH1^R132H^* mutant tumour implants [76].

### 6.4. C-C Motif Chemokine Ligand 18 (CCL18)/Chemokine (C-C Motif) Receptor 8 (CCR8)/Acid Phosphatase 5 (ACP5)/AKT1 Substrate 1 (AKT1S1/PRAS40)/Akt Pathway

CCL18 is a member of the CC chemokine family and is predominantly secreted by myeloid cells such as monocytes, macrophages and dendritic cells. Recent studies revealed that CCL18 plays a pivotal role in the epithelial-mesenchymal transition in pancreatic and breast cancer [126,127]. Huang et al. have used an ex vivo model of induced pluripotent stem cell (iPSC)-derived human microglia/macrophages and shown that the cells upregulate CCL18 and induce glioma cell growth and invasion via the CCR8/ACP5/AKT1S1 (PRAS40)/Akt pathway [82] (Figure 3, Molecular event 30). Importantly, an increased expression of CCL18 is inversely correlated with survival time in primary and recurrent *IDH*-WT GBM patients [82].

### 6.5. ATP Binding Cassette Subfamily A Member 1 (ABCA1)

Patients affected by *IDH1* mutant high-grade glioma and IDH-WT GBM, respectively, show distinct clinical features and prognostic differences. Wildtype IDH1 and IDH2 catalyse the conversion of isocitrate to alpha-ketoglutarate (α-KG), whereas the IDH1 and IDH2 mutant enzymes exhibits a neomorphic function catalysing the reduction of α-KG to oncometabolite D-2-hydroxyglutarate (D-2HG) [128]. In a hypoxic environment, the *IDH1*-dependent pathway is significantly up-regulated and facilitates reductive glutamine metabolism which is used in lipogenesis and maintains the proliferation of GBM cells [128,129]. By utilising annotation analysis of metabolism-related genes, Wang et al. suggested that macrophages in *IDH*-WT GBMs significantly increase expression of ABCA1 [130]. The *ABCA1* gene encodes a membrane-associated protein that uses cholesterol as its substrate and induces cholesterol efflux in the cellular lipid removal pathway [130,131,132]. Interestingly, down-regulation of *ABCA1* expression can restore a pro-inflammatory phenotype in tumour-associated macrophages and may provide a therapeutic target for *IDH*-WT GBM [130].

## 7. Exosomes, Extracellular Vesicles and MicroRNAs in Glioma Progression

Exosomes are small membrane vesicles that are crucial for intercellular communication [133]. Recently, tumour-derived exosomes have been found to play an essential role in the immunosuppressive effects on immune cells by delivering several types of proteins and non-coding RNAs such as miRNAs, long noncoding RNAs and circular RNAs (circRNAs) [133,134]. A number of studies are now suggesting that a hypoxic tumour environment can modify the genetic content within exosomes and modulate the cell functions of recipient cells [135,136].

### 7.1. Interleukin-6 (IL-6) and MicroRNA-155-3p (miR-155-3p)

Various cell types in the TME of cancers release IL-6, leading to the activation of the IL-6/JAK/STAT3 pathway in both tumour cells and tumour-associated immune cells, which in turn promotes tumour cell proliferation, invasiveness and metastasis [137]. MiR-155-3p is involved in tumorigenesis and is highly expressed in several cancers, including breast cancer and glioma [138,139]. A recent experimental study by Xu et al. [44] reported that exosomes derived from human GBM cell lines that had been subjected to hypoxia stimulated autophagy in macrophages (cell lines). Elevated IL-6 and miR-155-3p levels in the exosomes appeared to be responsible for the effect. Further Western blot analysis revealed that IL-6 triggers autophagy in macrophages by activating STAT3 signalling (Figure 2, Molecular event 15). Using flow cytometry and ELISA, the authors also found that increased IL-6 and miR-155-3p in the treated exosomes significantly induced CD163 and IL-10 expression in macrophages [44]. Taken together, IL-6 and miR-155-3p delivered by “hypoxic” exosomes derived from human GBM cells drive macrophages towards an immunosuppressive phenotype which supports glioma proliferation and migration.

### 7.2. MicroRNA-1246 (miR-1246)

Qian et al. [69] have demonstrated that miR-1246 contained within “hypoxic” glioma-derived exosomes (H-GDEs) is capable of inducing an immunosuppressive phenotype in macrophages (increased CD163, IL-10, IL1RA, TGFβ1 and CCL2 [140] expression, and significantly decreased TNF-α expression) [69]. In addition to its immunosuppressive function, further analysis revealed that miR-1246 also binds to the 3′-untranslated region of Telomeric repeat-binding factor 2-interacting protein 1 (TERF2IP) leading to inhibition of its expression as well as induction of an immunosuppressive phenotype in macrophages via STAT3 and NF-κB [69] (Figure 2, Molecular event 13). Activation of STAT3 and inhibition of the NF-κB pathway which foster an immunosuppressive TME also promote the proliferation, migration and invasion of glioma cells in vitro as well as in vivo [141]. Interestingly, abundant expression of miR-1246 is found in the CSF (cerebrospinal fluid) of GBM patients and significantly reduced following surgical resection [69].

### 7.3. Circ_0012381, Arginine Deprivation

Since radiated GBM cells release exosomal circ_0012381 which induces M2 polarization of microglia leading to better growth of GBM via the CCL2/CCR2 axis, Zhang et al. [142] suggested that inhibition of exosome secretion might represent a potential therapeutic strategy to improve the efficacy of radiotherapy in GBM patients. Arginine deprivation may also have a positive therapeutic effect in a subset of cases [143].

### 7.4. MicroRNA as a Potential Therapeutic Tool for Targeting Glioma

Extracellular vesicles (EVs) are released by GSCs and used for their communication with microglial cells and brain macrophages. For instance, metastasis-associated lung adenocarcinoma transcript 1 (MALAT1) modulates the inflammatory response of microglia after LPS stimulation through regulating the miR-129-5p/HMGB1 axis [144]. Exposure of microglia to EVs released by hyperbaric oxygen-treated GBM cells up-regulates the expression of pro-inflammatory cytokines IL-1β, IL-6 and STAT1 and down-regulates the anti-inflammatory cytokine PPARγ [145]. This finding was further corroborated by a recent study by Wang et al. [146] who proposed that via EVs, Cavin1 overexpressing glioma cells exert a general activating effect on microglia/macrophages. In turn, microglia-derived EVs modify tumour cell metabolism and enhance glutamate clearance through miR-124 leading to reduced glioma growth [147]. These findings are significant because microRNA-loaded EVs have been proposed as the basis for a new type of glioma therapy as miR-124 delivery exerts synergistic anti-tumour effects by inhibiting M2 microglial polarization and suppressing the growth of human GBM [148] (cf. “Janus” genes below and Table 1).

## 8. Effects of Epigenetic Modifications in Glioblastoma

Epigenetic mechanisms transform transient signals into persistent cellular responses, e.g., via DNA methylation, acetylation and phosphorylation [149]. These epigenetic modifications are carried out by histone and chromatin modifiers [150]. Acetylation of the N-terminal lysine residues at histone H3 and H4 is associated with unpacking of chromatin to activate gene transcription. In contrast, methylation of H3 lysine 9 (H3K9me3) and 27 (H3K27me3) are hallmarks of compacted chromatin and gene silencing [149].

### 8.1. Histone Deacetylases (HDAC)5 and HDAC9 and “Trained Microglia Immunity”

A recent study has shown that glioma-conditioned medium (GCM) induces microglia to acquire a modified transcriptional signature characterized by reduced expression of pro-inflammatory genes (*c-Myc*, *Mark1*) following sequential exposure to lipopolysaccharide (LPS) [78]. Such microglia demonstrate increased histone deacetylase, HDAC5 and HDAC9 activities and repressed histone trimethylation (H3K27me3) at inflammatory genes (*iNOS*, *Zbp1*, or *Irf7*) after GCM exposure [78] (Figure 3, Molecular event 26). It is worth noting that HDAC inhibitors can erase glioma-induced epigenetic modifications and glioma-polarised microglia are able to re-establish their ability to activate pro-inflammatory genes [78].

### 8.2. Histone H4 Lysine 16 (H4K16) Acetylation

Glioma-induced conversion of microglia into glioma supportive cells is associated with an increase of H4K16 acetylation in microglia and augmented nuclear relocation of the deacetylase SIRT1, which in turn stimulates deacetylation of the H4K16 acetyltransferase hMOF and its recruitment to promoter regions of genes involved in microglia activation (*Ccl22, Chil3, Il6, Mmp14*) [151].

### 8.3. Enhancer of Zeste Homolog 2 (EZH2) Silencing

The H3K27M mutation is an intrinsic hallmark of H3K27-altered paediatric diffuse midline glioma [88]. Recent studies have suggested that the H3-K27M mutation inhibits the activity of histone-lysine N-methyltransferase Enhancer of zeste homolog 2 (EZH2), a catalytic subunit within the polycomb repressive complex 2 (PRC2) [152]. It is known that PRC2 plays a crucial role in the maintenance of transcriptional silencing via trimethylation of H3K27 (H3K27me3) [152,153]. Recently, Keane and colleagues [154] were able to show that silencing of *EZH2* in BV2 microglia results in significantly increased phagocytosis of diffuse midline glioma cells. However, inhibition of EZH2 had minimal impact on the growth of tumour cells [154]. In other words, EZH2 inhibition in microglia rather than tumour cells has antitumoral effects in diffuse midline glioma [154]. Importantly, it had been previously shown that EZH2 suppression in GBM can rescue microglia immune functions [155].

### 8.4. Interferon Regulatory Factor 8 (Irf8) Promoter Demethylation

Using transplantation of specifically engineered mesenchymal GSCs into immunocompetent mice, Gangaso et al. [74] discovered that mesenchymal GSCs are capable of mimicking the activation of myeloid-specific genes through *epigenetic immunoediting* (EI). EI refers to a process where through exposure to an in vivo environment followed by immune attack, GSCs undergo site-specific DNA methylation changes alongside concomitant transcriptional changes that lead to activation of several immune-related ‘signatures’. For instance, *Irf8*, a myeloid-specific master transcription factor, is normally exclusively expressed in hematopoietic cells [156,157] and has known specific roles in the differentiation of myeloid lineages [158,159]. Strikingly, the mesenchymal GSCs were observed to activate *Irf8* by progressively erasing methylation in the *Irf8* promoter region and gene body in response to sustained IFNγ stimulation [74]. Of note, IFNγ secreted by infiltrating macrophages may stabilise activation of Irf8 in GSCs and drive maturation of myeloid cells to further produce IFNγ [160] (Figure 3, Molecular event 21).

## 9. The Importance of Signal Transducer and Activator of Transcription 3 (STAT3) in the Tumor Microenvironment

STAT3 is central to GBM pathology as it links multiple pathways that are important for the creation of this specific tumour microenvironment (TME). For instance, microglia/macrophage-derived IL-1β supports GBM growth via the STAT3/NF-κB pathway [161].

### 9.1. Anti-Inflammatory Phenotype of GBM-Associated Reactive Astrocytes

Heiland et al. [83] have reported that a distinct anti-inflammatory phenotype of reactive astrocytes in GBM is associated with activation of the JAK/STAT pathway and increased CD274 (PD-L1) expression [83]. RNAseq analysis revealed that this reactive phenotype of glioma-associated astrocytes is induced by microglia in the TME [83]. Interestingly, tumour-associated astrocytes may potentially mediate specific re-programming of gene transcription in microglia as a small subset of genes (*APOE, APOC2, HLA-DRA*) was exclusively up-regulated in microglia in the neighbourhood of TME astrocytes (Figure 3, Molecular event 33). It is noteworthy that microglia also exhibited an accelerated hypoxic metabolism characterised by increased glycolytic activity when co-cultured with reactive astrocytes of the TME [83]. As predicted, pharmacological silencing of the JAK/STAT pathway using Ruxolitinib shifted the TME towards a pro-inflammatory state [83].

### 9.2. GSC-Derived Exosomes Contain Immunosuppressive Molecules of the STAT3 Pathway

Gabrusiewicz and colleagues [72] were able to show that glioblastoma stem cell (GSC)-derived exosomes have an affinity for CD11b expressing monocytes. CD11b is a key microglia marker. Importantly, additional mass spectrometry studies revealed that GSC-derived exosomes contain various immunosuppressive molecules including members of the STAT3 pathway, such as Akt, Erk1/2 and mTOR [72]. Furthermore, the authors also found that GSC-derived exosomes can induce CD163 in monocytes [72] (Figure 2, Molecular event 17).

### 9.3. STAT3-MYC Proto-Oncogene, BHLH Transcription Factor (MYC) Activation Induced Temozolomide (TMZ) Resistance

A recent study by Li et al. [80] has pointed out that GBM patients that respond exceptionally well to TMZ typically show fewer microglia/macrophages in association with GBM. Moreover, glioblastoma associated microglia/macrophages up-regulate expression of the proinflammatory cytokine interleukin 11 (IL-11), resulting in activation of the STAT3-MYC pathway in GBM cells. In addition to activating the STAT3 pathway, xenograft studies have shown that this mechanism is responsible for the induction of a cancer stem cell state in GBM cells (OLIG2, SOX2, and POU3F2 are up-regulated) [162]) leading to higher tumorigenicity and TMZ resistance (Figure 3, Molecular event 28). Furthermore, the production of microglial IL-11 was shown to depend on activation of the gamma isoform of myeloid-specific phosphoinositide-3-kinase (PI3Kγ) [80]. Using orthotropic murine GBM models, the authors demonstrated that inactivation of PI3Kγ reduces expression of microglia/macrophage-derived IL-11 and enhances TMZ efficacy. When STAT3 is silenced, migration of cancer stem cells towards macrophage secreted factors appears to be reduced; in fact, co-habitation of glioma stem cells and macrophages appears to result in bi-directional signalling that alters the phenotypes of both cell types [163].

## 10. Sources of Proangiogenic Factors in Glioma

### 10.1. CircKIF18A-Forkhead Box C2 (FOXC2) Complex

Accumulating evidence indicates that exosomes are involved in the tumorigenesis, angiogenesis and progression of GBM. Exosomes in the TME enable transferring of biological molecules and intercellular communication among GBM cells, GBM-associated immune cells and endothelial cells [164]. An experimental study by Jiang et al. [67] has uncovered that microglia exhibiting an immunosuppressive phenotype marked by up-regulation of CD163, MRC1 (CD206) [165], ARG1, IL-6 [43,161] and TGF-β and down-regulation of IL-1β and TNF-α are capable of stimulating the migration, invasion and tube formation of human brain microvessel endothelial cells (hBMECs) in vitro and in vivo. In addition to microglia, GBM-associated endothelial cells are another important source of IL-6 in the TME [43]. Additional experiments showed that microglia are able to induce angiogenesis in GBM by transporting circKIF18A to hBMECs via microglia-derived exosomes. Exosomal circKIF18A then binds to FOXC2 in the cytoplasm of hBMECs and induces translocation of FOXC2 to the nucleus. Accordingly, CircKIF18A-FOXC2 complex up-regulates the expression of ITGB3, CXCR4, and DLL4 in hBMECs. It is worth noting that FOXC2 binding to circKIF18A can concomitantly activate the PI3K/AKT pathway and further promote angiogenesis and GBM growth [67] (Figure 2, Molecular event 11). Moreover, FTY720, a potent immunosuppressant, may be potentially used to inhibit glioma growth by inhibiting MAPK-mediated secretion of IL-6 following increased internalization of CXCR4 [166].

### 10.2. Yes1 Associated Transcriptional Regulator (YAP1)/Lysyl Oxidase (LOX)/β1 Integrin (ITGB1)/Protein Tyrosine Kinase 2 (PYK2) Pathway

It is now evident that the mesenchymal subtype of GBM is enriched for *PTEN* and *NF1* mutations [167]. Recently, Chen et al. [36] have reported that GBM cells lacking *PTEN* stimulate BMDM infiltration via activation of the YAP1/*LOX*/ITGB1/PYK2 pathway. The infiltrated BMDMs secrete Spp1 which sustains glioma cell survival by inhibiting apoptosis and augmenting angiogenesis [36] (Figure 3, Molecular event 19). The Hippo-YAP1 signalling pathway, a regulator of cell proliferation and stem cell functions, is known to play a pivotal role in cancer progression. YAP1 is a key mediator of *LOX* transcription in GBM [168].

### 10.3. Advanced Glycosylation End-Product Specific Receptor (AGER/RAGE)

Under homeostatic physiological conditions, high expressions of AGER (RAGE) has only been observed in the lung, and its levels are low in other cell types. However, during chronic inflammation and in neurodegenerative diseases, significant levels of AGER can be observed in activated endothelial cells [169]. More recently, the expression of AGER in tumour-associated microglia and macrophages has been shown to promote angiogenesis in glioma suggesting that targeting the interaction of AGER ligands with their receptors may have therapeutic potential [170].

### 10.4. Polymorphonuclear Leukocytes/Granulocytes (GBM-hPMNL)

An alternative source of proangiogenic factors in glioma has been identified by Blank et al. [27]. These authors demonstrated that GBM which contain a higher number of polymorphonuclear leukocytes/granulocytes (GBM-hPMNL) are also characterised by increased levels of CD163, TEK (TIE2), HIF1α, VEGF, CXCL2 and ANPEP (CD13). Using double-labelling for immunofluorescence microscopy, the authors observed that CD163 and TEK (TIE2) were exclusively expressed by AIF1 (Iba1)+ cells in GBM-hPMNL specimens. Further analyses revealed up-regulation of CXCL2, IL-8 and ANPEP (CD13) which are considered alternative proangiogenic factors [27] (Figure 1, Molecular event 8). A previous study by the same group had shown that CXCL2 can promote angiogenesis independently of the classical angiogenic molecule VEGF [38]. Of note, CXCL2, which has been previously shown to attract granulocytes in mice [25], was found to be highly unregulated by both glioma cells and microglia/macrophages. Granulocytes were found to be the main source of IL-8 in GBM [26] which is known to induce recruitment of macrophages [28]. Interestingly, more than 50% of the tumour blood vessels appeared to have interactions with AIF1+ cells and some were completely surrounded by microglia/macrophages [27]. Taken together, interactions between microglia/macrophages and granulocytes may play an important role in remodelling and stabilising glioma vasculature for efficient oxygen supply to GBM tissue.

## 11. Molecules Promoting Survival of Glioma Stem Cells

Recent studies suggest that GSCs and TAMs co-reside in hypoxic and perivascular niches and play pivotal roles in the recurrence of GBM after radiation and chemotherapy. The dynamic communication between GSCs and TAMs suggests a functioning relationship in support of the malignant growth of GBM [9,24].

### 11.1. Serrate RNA Effector Molecule Homolog (SRRT/ARS2)/Monoacylglycerol Lipase (MGLL)/Prostaglandin E2 (PGE2)/β-Catenin Pathway

SRRT (ARS2) is known to have a crucial role in the regulation of cell proliferation and mammalian development [171]. It is a key transcription factor involved in the self-renewal of neural stem cells (NSCs) and controls the multipotent progenitor state of NSCs by directly activating the pluripotent gene, *SOX2* [172]. Yin et al. [73] recently reported that SRRT regulates the stem cell-like properties (nestin expression and sphere-forming ability) of GSCs by directly inducing the expression of MGLL (MAGL) via activation of transcription of the *MGLL* gene. Moreover, MGLL is able to promote self-renewal and increase tumorigenicity of GSCs by up-regulating the secretion of PGE2. PGE2 and PGE2 mediated β-catenin induce an immunosuppressive phenotype in macrophages by up-regulating the expression of MRC1 (CD206), CD163, arginase-1 (ARG-1) and Krupple-like factor 4 (KLF4) while down-regulating the expression of TNF-α and CD86 [73] (Figure 3, Molecular event 18). Accordingly, experimental pharmacological silencing of MGLL in mouse xenografts revealed that inhibition of MGLL and reduction of PGE2 expression down-regulates the stemness marker, nestin and up-regulates expression of the glial fibrillary acidic protein (GFAP) [73]. As expected, glioma xenograft mice with silenced MGLL showed significantly prolonged survival [73].

### 11.2. Clock Circadian Regulator (CLOCK)/Basic Helix-Loop-Helix ARNT Like 1 (BMAL1)/Olfactomedin Like 3 (OLFML3) Feed-Forward Loop

It is now evident that circadian rhythm plays an important role in cancer biology affecting tumour cell metabolism, DNA repair and proliferation [173]. Of note, the CLOCK-BMAL1 complex influences transcription and can exhibit a pro- or anti-oncogenic role depending on cues received from the TME [173]. It has been suggested that CLOCK, which is amplified in about 5% of GBMs, and BMAL1 are involved in the recruitment of microglia (via OLFML3) and GSC renewal [39]. Strikingly, microglia also express OLFML3 [40], and Neidert et al. [40] and Chen et al. have suggested that microglia may stimulate the recruitment of more microglial cells (feed-forward loop) [39] (Figure 3, Molecular event 22). In line with these results, a more recent study has shown that microglia derived OLFML3 acts as a paracrine factor that facilitates glioma invasion [42]. Moreover, this idea is further supported by the recent finding that microglial exosomes and miRNA have been found to induce glioma progression by regulating circadian genes [41]. In turn, Dong et al. have proposed to target GSCs through disruption of their circadian clock [174].

### 11.3. Pleiotrophin (PTN)-Protein Tyrosine Phosphatase Receptor Type Z1 (PTPRZ1)

Accumulating evidence indicates that binding of PTN to PTPRZ1 promotes cell survival, adhesion and migration by inducing the phosphorylation of downstream tyrosine receptors [175]. Shi et al. [24] have recently observed that CD163+ glioma-associated macrophages produce large amounts of PTN which supports self-renewal and maintenance of GSCs via PTN-PTPRZ1 paracrine signalling (Figure 1, Molecular event 6). Interestingly, the PTN receptor PTPRZ1 is predominately expressed by SOX2-positive GSCs and increased expression of PTPRZ1 is negatively correlated with overall survival in GBM patients [24]. Conversely, disrupting PTN expression in macrophages reduces the number of SOX2+ glioma cells in GBM bearing mice. Furthermore, blockade of PTPRZ1 via short hairpin RNA or anti-PTPRZ1 antibody significantly reduces growth of GBM and prolongs survival of GBM xenografts [24].

## 12. Factors That Contribute to the Mesenchymal Transition of Malignant Glioma

Compared to other GBM subtypes, mesenchymal GBMs display the highest percentage of microglia/brain macrophage infiltration [60].

### 12.1. Macrophage Receptor with Collagenous Structure (MARCO)

MARCO is a class-A scavenger receptor which is involved in fundamental macrophage functions such as phagocytosis and their role in inflammation [79]. In a recent study analysing the transcriptomic networks of tumour-associated macrophages, *MARCO* was identified as one of the most highly expressed genes that regulate the mesenchymal transition in GSCs [79]. The authors further describe that non-mesenchymal GSCs, when treated with MARCO^high^ TAM-derived conditioned media, demonstrated a drastically increased expression of mesenchymal-, stemness-, and invasion-associated factors (CD44, homeobox protein NANOG (NANOG), leukemia inhibitory factor (LIF), and matrix metallopeptidase 2 (MMP-2)) [111,176,177,178]). In addition, studies of patient derived GBM xenografts revealed that non-mesenchymal GSCs co-injected with MARCO^high^ TAMs exhibited a more aggressive phenotype. Furthermore, these GSCs were characterized by expression of key mesenchymal molecules (CD44, chitinase-3-like protein 1 (CHI3L1), and topoisomerase II alpha (TOP2A) as well as mesenchymal hallmarks such as increased invasive and stem-like cellular properties including resistance to ionising radiation [79]) (Figure 3, Molecular event 27).

### 12.2. CD44 Molecule

Recent studies have suggested that high expression of CD44 in glioma cells is associated with the mesenchymal transition, increased recruitment of macrophages and reduced survival time in glioma patients [179]. Based on CIBERSORT analysis, Du and colleagues [180] have proposed that CD44+ glioma cells communicate with pro-inflammatory microglia-derived brain macrophages through SPP1-CD44 signalling. Interestingly, tumour-associated astrocytes increase the migratory ability of glioma cells via up-regulation of glioma cell CD44. They also promote recruitment of pro-inflammatory macrophages which may play a role in inducing glioma stemness via the SPP1-CD44 pathway [180]. This observation is in line with earlier findings showing that high expression of *Gpnmb* and *Spp1* is detected in both murine and human glioma-associated microglia/macrophages and is associated with reduced survival time in patients with GBM [181]. Furthermore, Ivanova and colleagues [182] have observed that deletion of CD44 hinders invasion of glioma cells into the surrounding brain tissue. Using *CD44* knockout mice, the authors demonstrated that microglial CD44 plays an important role in the activation of TLR2 pathways and the regulation of MMP9 expression. Additional experiments revealed that CD44 deletion suppresses mRNA expression of TNF-α, IL-1β and MMP9 in microglia in both in vivo and in vitro settings. Moreover, a recent study proposed that ortho-vanillin (O-Vanillin) may be used to inhibit TLR2 mediated expression of MMP9, MMP14 and IL-6 [183]. CD133+ GSCs initiate microglial IL-6 secretion via TLR4 signalling [184]. Taken together, these studies provide important confirmation that communication between tumour-associated macrophages and GSCs is bi-directional.

### 12.3. Macrophage-Secreted Oncostatin M (OSM)

Employing scRNA-seq and functional experiments, Hara et al. [185] were able to show that glioma-associated macrophages are capable of inducing “mesenchymal-like states” in GBM cells by releasing an epithelial-to-mesenchymal transition inducer, OSM. Macrophage-derived OSM binds to the cognate OSM receptor expressed by GBM cells and mediates downstream STAT3 signalling [185]. Furthermore, other genes associated with mesenchymal-like states are *VIM, CD44,* and *ANXA1*. Interestingly, mesenchymal-like states in TAMs are associated with increased expression of cytotoxicity markers in T-cells (e.g., *GZMB* and *PRF1*) and up-regulation of MHC I and MHC II genes [185].

### 12.4. Fibrinogen-like Protein 2 (FGL2)/Fc Gamma Receptor III (CD16)/PI3K/Akt/Hypoxia Inducible Factor 1 Subunit Alpha (HIF1α)/Pro-Platelet Basic Protein (PPBP/CXCL7) Paracrine Loop

High levels of FGL2 in GBM have been shown to induce immunosuppression through up-regulation of programmed cell death protein 1 (PD-1) and ectonucleoside triphosphate diphosphohydrolase 1, also known as CD39 [186]. It has been shown that FGL2-mediated immunosuppression plays an essential role in the malignant progression of GBM and in modulating the transformation of low-grade astrocytoma to GBM [186]. A subsequent study by the same authors [75] has shown that glioma-derived FGL2 induces secretion of PPBP (CXCL7) by a subset of TAMs following activation of FGL2/CD16/PI3K/Akt/HIF1α signalling, which promotes the expression of stem-like properties in glioma cells (Figure 3, Molecular event 23). As predicted, disruption of the FGL2/PPBP paracrine loop through *FGL2* knockout and anti-PPBP antibody binding significantly suppresses the tumorigenic effects of FGL2 and prolongs survival in tumour-bearing mice [75].

### 12.5. MicroRNA-504 (miRNA-504)

Emerging evidence suggests that certain miRNAs have a prominent role in gliomagenesis [187]. MiR-504 appears to be another “Janus” factor (please see following section and Table 1). When over-expressed in GSCs it not only inhibits their tumorigenic potential but can be also passed on to microglia where miR-504 promotes M1 polarization. Specifically, miR-504 modulates the stemness and mesenchymal transition of glioma stem cells and their interaction with microglia via delivery in extracellular vesicles [45].

## 13. Synergistic (“Janus”) Genes and Pathways That Act in Both Glioblastoma Cells and Microglia/Macrophages Supporting Glioblastoma Progression

### 13.1. Cellular Communication Network Factor 4 (CCN4)/Integrin α6β1 (ITGα6β1)/Akt Pathway

Wnt/β-catenin is well-known to be involved in modulating cell proliferation, migration and apoptosis and is also known to play a critical role in cancer progression [188]. In GBMs, the Wnt/β-catenin pathway is aberrantly activated in GSCs and drives malignant tumour progression [189]. A recent study by Tao and colleagues [32] has demonstrated that CCN4, also known as Wnt1-inducible signaling pathway protein-1 (WISP1) is preferentially released by GSCs leading to a tumour-supportive cellular environment which fosters survival of both GSCs and tumour-associated macrophages. In addition, CCN4 facilitates self-renewal and proliferation of GSCs by binding to Integrin α6β1 (ITGα6β1) and activating the Akt pathway in an autocrine manner. Strikingly, the survival of tumour-supportive macrophages is also facilitated by the CCN4/ITGα6β1/Akt pathway but via a paracrine signalling loop [32] (Figure 3, Molecular event 32). A xenograft study by the same authors further revealed that inhibition of the Wnt/β-catenin/CCN4 pathway disrupts the growth of GSCs and induces apoptosis of tumour-associated macrophages [32].

### 13.2. Chitinase-3-like Protein 1 (CHI3L1/YKL-40)/Galectin 3 (Gal3)/PI3K/Akt/mTOR Pathway

CHI3L1 (YKL-40) is a secreted glycoprotein that belongs to the glycoside hydrolase family 18 and plays a role in tissue remodelling [190]. Results of a recent study indicate that CHI3L1 promotes tumorigenesis in GSCs that lack MGMT methylation which is relevant for TMZ resistance [191]. Mechanistically, Galectin 3 (Gal3) binds to CHI3L1, and is encoded by the *LGALS3* gene. Furthermore, formation of the CHI3L1-Gal3 protein complex drives macrophages towards an immune suppressive phenotype [33]. Strikingly, the protein complex formed by CHI3L1-Gal3 promotes the infiltration of “pro-tumour” M2 but not “anti-tumour” M1 macrophages and appears to be regulated transcriptionally by NF-κB/CEBPβ in the CHI3L1/Gal3/PI3K/Akt/mTOR axis [33]. It is worth noting that CHI3L1 is also expressed in GSCs and regulated by the PI3K/Akt/mTOR pathway in a positive feedback loop (Figure 3, Molecular event 31) [33].

## 14. Mechanisms Underlying Therapy Resistance of GBM

This is a frequently discussed topic. We would like to focus on aspects that are less commonly discussed in the literature but may deserve additional attention.

### 14.1. Long Noncoding RNAs (lnc-TALC)

A recent study by Li et al. [70] has revealed that long noncoding RNAs which are induced by TMZ-treatment in recurrent GBM (lnc-TALC) are able to promote TMZ resistance by transporting Inc-TALC to microglia via glioma derived-exosomes (GDEs). In addition, increased levels of Inc-TALC in microglia induce an immunosuppressive phenotype by stimulating the secretion of TGF-β, IL-4, and IL-10 [70]. Mechanistically, exosomal Inc-TALC induces activation of the p38 MAPK pathway in microglia by binding to cytoplasmic enolase 1 (ENO1) which in turn up-regulates the secretion of complement components C5/C5a [70]. A previous study carried out by the same group reported that C5/C5a released from glioma-associated microglia binds to the C5a receptor (C5aR) on glioma cells and induces TMZ-resistance by elevating the expression of DNA damage repair (DDR)-related proteins [71] (Figure 2, Molecular event 14). On the flip side, blockade of lnc-TALC–mediated communications between microglia and glioma cells led to increased TMZ sensitivity and prolonged lifespan in a mouse GBM model [71]. Importantly, a recent study has found that TMZ resistance can be overcome by modulation of the lncRNA SNHG15/CDK6/miR-627 circuit by Palbociclib which also reduces M2-polarization of microglia in glioma [192]. In experimental animals, survival of TMZ-resistant glioma bearing mice can be improved by combination therapy with a p38 MAPK inhibitor and PD-L1 antibody and this seems to work via a reduction of infiltrating glioma-associated macrophages and PD-L1 expression on resident glioma-associated microglia [193]. Moreover, glioma cells can be sensitised to TMZ by silencing Metadherin (MTDH/AEG-1) which also attenuates M2-polarisation of glioma-associated microglia/macrophages [194].

### 14.2. Colony-Stimulating Factor-1 Receptor (CSF-1R)

The CSF family of cytokines has been a focus of attention in glioma research over the last decade [195,196]. A recent study employing mouse models reported that inhibition of CSF-1R has potential to be used clinically to counter radiation resistance. In an irradiated TME, blockage of CSF-1R hinders the induction of SMAD and RBPJ which are members of the TGF-β and Notch pathway, respectively, and which have a role in the expression of microglia and BMDM immunosuppressive phenotypes [197,198,199,200] appears to be the underlying mechanism. Following CSF-1R inhibition, microglia and BMDM may not interfere with the benefits of initial radiotherapeutic tumour-debulking that leads to a significant delay in glioma recurrence [201]. Moreover, the work of Rao et al. suggests that CSF1R inhibition blocks the growth of PDGFB-overexpressing glioma, e.g., proneural glioma with PDGFB over-expression whereas mesenchymal GBMs are resistant to this approach [202]. In other words, the TME differs between GBM subtypes and so does TAM function and responsiveness to CSF1R inhibition [203]. Furthermore, targeting the EGFR ligand amphiregulin might counteract the microglial stimulation of glioma invasion [204].

### 14.3. 5′- Nucleotidase Ecto (NT5E)

NT5E is an ectonucleotidase that is well known as 5′-nucleotidase in brain research and functions with upstream CD39 to convert extracellular ATP into adenosine [205]. In a recent experimental study by Goswami et al. [206] it was reported that GBM contain a subset of CD68+ macrophages that co-express NT5E (CD73) and survive anti-PD-1 treatment. Further analysis revealed that macrophages expressing high levels of NT5E are characterized by elevated expression of chemokines and chemokine receptors such as *CCR5, CCR2, ITGAV/ITGB5* and *CSF1R*, which may play a role in the recruitment of BMDM into the GBM microenvironment [206]. Importantly, silencing of NT5E impedes intracranial tumour growth. Moreover, NT5E-/- mice with GBM exhibit prolonged survival following treatment with anti-CTLA-4 and anti-PD-1 suggesting that NT5E may be a causative factor in failures of the immunotherapy of GBM [206]. Interestingly, concomitant up-regulation of iNOS in myeloid cells was observed in *NT5E* knockout mice [206].

### 14.4. Mechanistic Target of Rapamycin Kinase (mTOR)

The mTOR pathway is known to promote cell proliferation and survival in various cancers, and its expression is also up-regulated in GBM [207]. Since clinical trials targeting the mTOR pathway in GBM have not shown the expected results so far, it is of great interest that microglia rather than tumour cells might be used as the primary target of mTOR inhibition in GBM [208]. Recently, Dumas et al. found that microglia promote GBM via mTOR-mediated immunosuppression of the TME [208]. The authors further demonstrated that GBM-initiating cells induce mTOR signalling in microglial cells but not bone marrow-derived macrophages [208].

## 15. Recent Advances in the Development of Novel Therapeutic Approaches to Treat GBM

It is becoming increasingly clear that the diversity of immune cell proportions and phenotypes within glioma are determined by glioma cells and their molecular features [209]. Martins et al. have recently proposed regimens that locally modulate tumour-associated microglia to achieve long-lasting and effective tumoricidal responses [210]. Furthermore, a nucleic acid nanogel that mimics surface properties of the influenza virus has been used to reprogram microglia and macrophages for GBM therapy [211]. This is particularly interesting because tumour-associated microglia and macrophages often contribute to the failure of oncolytic virotherapy by preventing an efficient intratumoral viral distribution [212]. Another study has suggested that engineered microglia can be used to potentiate the action of drugs against glioma [213]. In the latter study extracellular vesicles and tunnelling nanotubes were employed. Recently, Mormino and colleagues genetically modified microglia to secrete IL-15 which enhances the recruitment of NK cells and helps maintain a pro-inflammatory phenotype in microglia and NK cells that hinders glioma growth [214]. In addition, the group of Yong found [215] that amphotericin B can be used to activate monocytoid cells which subsequently overcome tumour-induced immune suppression in glioma and suppress brain tumour-initiating cells.

## 16. Conclusions

At the turn of the millennium the finding that microglia support glioma growth seemed counterintuitive. Neuro-oncology at the time was dominated by molecular geneticists with a sole focus on the neoplastic cell. The microenvironment of a glial tumour was of little interest in comparison. A decade later, irrefutable evidence had accumulated that gliomas manipulate the functions of microglial cells. It also became clear that tumour-associated brain macrophages are derived from different sources, notably bone marrow in addition to local microglia. Feedforward another 10 years, it is now clear that the tumour microenvironment likely holds the key to effective glioblastoma therapy. Tumour-associated microglia and macrophages can now be considered regulators of malignancy in glioma [215]. Exosomes, extracellular vesicles, microRNAs and epigenetic modifications clearly have a role in glioma progression. The discovery of synergistic, or as we propose to call them, “Janus” genes and pathways that act in both glioblastoma cells and microglia/macrophages supporting glioblastoma progression may provide exciting new therapeutic opportunities. Malignant glioma cells/GSCs and microglia/macrophages have an intricate relationship and targeting several synergistic mechanisms at once seems particularly attractive. Moreover, modifying microglia directly for the treatment of glioma may provide novel access to glioblastoma and other high grade gliomas [216]. The therapeutic potential of targeting tumour associated microglia/brain macrophages has been the topic of a number of recent review articles [217,218,219]. Expression of certain glioma- and TAM-derived pro-inflammatory molecules provides malignant glioma with alternative stimuli for growth and therapy resistance. For instance, IFNγ stabilises *Irf8* activation in glioma cells and microglial C5/C5a complement component induced TMZ resistance [70,74]. Glioma cell-TAM fusion products [119] might represent the ultimate “Janus” mechanism providing glioma cells with additional migration and infiltration properties under macrophages camouflage. Interestingly, microglial features have been described in a case of epithelioid GBM which is a rare variant of GBM that typically occurs in children and young adults [220]. “Janus” factors and pathways together with their positive feedback loops appear to represent key mechanisms that allow glioblastoma to growth so successfully. Recent advances in understanding the functions of non-coding RNAs may also offer novel therapeutic insights. New approaches may include modifying the sequence of non-coding RNAs and generating levels of non-coding RNAs that can trigger a pro-inflammatory phenotype in TAMs [192]. 

## Figures and Tables

**Figure 1 ijms-23-15612-f001:**
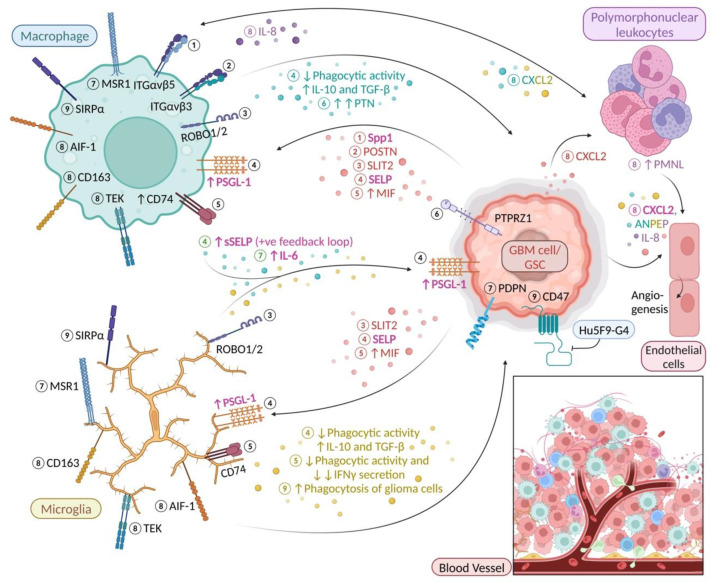
Microglia and macrophage surface molecules and cytokines in glioma. Molecular event ① Glioma cell-derived Secreted phosphoprotein 1 (Spp1) binds to ITGαvβ5 and induces macrophage recruitment to the TME [19] ② Glioma cell-derived periostin (POSTN) binds to ITGαvβ3 stimulating BMDM to enter the TME from the bloodstream [20] ③ GBM-derived Slit guidance ligand 2 (SLIT2) binding to Roundabout 1 and 2 (ROBO1/2) mediates microglia/macrophage chemotaxis [21] ④ GBM-derived SELP interacts with PSGL-1 altering microglia/macrophage phenotype by reducing phagocytic activity while upregulating IL-10 and TGF-β; exposure to soluble SELP (sSELP) creates a positive feedback loop leading to overexpression of P-selectin (SELP) and P-selectin glycoprotein ligand-1 (PSGL-1) by GBM and microglia cells [22] ⑤ Cells belonging to the monocytic subset of myeloid-derived suppressor cells (M-MDSCs) express high levels of CD74 in the presence of glioma-derived Macrophage migration inhibitory factor (MIF); CD74 activation weakens the microglial defence against glioma cells [23] ⑥ CD163+ glioma-associated macrophage secreted pleiotrophin (PTN) supports self-renewal and maintenance of GSCs via PTN–Protein tyrosine phosphatase receptor type Z1 (PTPRZ1) paracrine signalling [24] ⑦ MSR1+ microglia/macrophages in GBM upregulate expression of IL-6 and often co-reside with stem-like GBM cells expressing PDPN [18] ⑧ CD163 and TEK (TIE2) are exclusively expressed by AIF1 (Iba1)+ cells in GBM which contain a higher number of polymorphonuclear leukocytes/granulocytes; CXCL2 has been previously shown to attract granulocytes in mice and was found to be highly up-regulated by both glioma cells (red) and microglia/macrophages (yellow and blue) [25]; granulocytes were found to be the main source of IL-8 in GBM (purple) [26] stimulating recruitment of macrophages; CXCL2, IL-8 and ANPEP (CD13; blue, yellow and purple) are alternative proangiogenic factors [25,26,27,28] ⑨ anti-CD47 antibody Hu5F9-G4 promotes microglia/BMDM-mediated phagocytosis of glioma cells by blocking CD47-SIRPα interaction [29,30,31]. Font colour: magenta, “Janus” factors that are secreted by both microglia/macrophages and GBM cells/GSCs (cf. Table 1); green, molecules secreted by both microglia and BMDMs. “Created with BioRender.com”.

**Figure 2 ijms-23-15612-f002:**
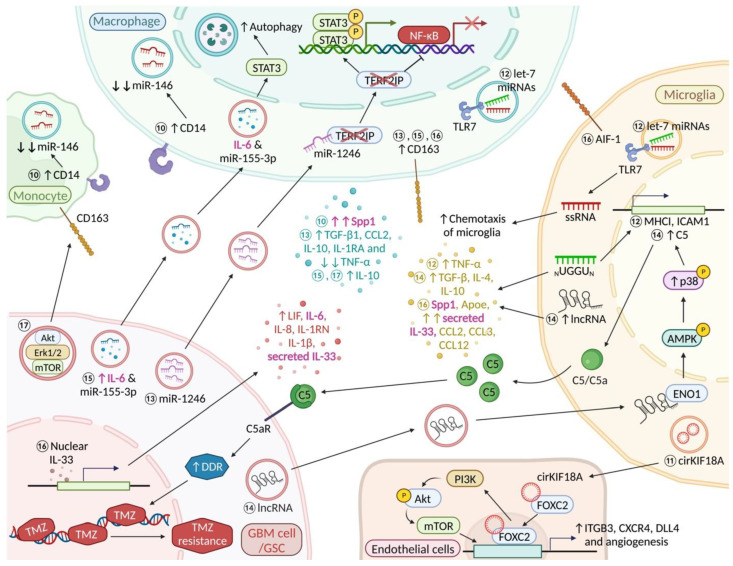
MicroRNAs, exosomes and a nuclear cytokine in microglia/macrophage-glioma interactions. Molecular event ⑩ Tumour supportive CD14+ monocytes and BMDMs downregulate microRNA-146 (miRNA-146) expression; CD14+ BMDM express high levels of Spp1 [35] ⑪ Microglial circKIF18A induces angiogenesis via CircKIF18A-FOXC2 nuclear translocation and *ITGB3, CXCR4*, and *DLL4* transcription while concomitantly activating the PI3K/Akt pathway further promoting angiogenesis and GBM growth [67] ⑫ Let-7 microRNAs containing the sequence motif UUGU can suppress glioma growth through upregulation of TNF-α, MHC I and ICAM1; miRNA oligoribonucleotides that lack the GU-rich core motif may act as a chemoattractant for microglia in glioma [68] ⑬ miR-1246 binding to Telomeric repeat-binding factor 2-interacting protein 1 (TERF2IP) stimulates an immune suppressive phenotype in macrophages (increased CD163, IL-10, IL1RA, TGFβ1 and CCL2 expression, and significantly decreased TNF-α) via STAT3 and NF-κB pathways [69] ⑭ GBM-derived long noncoding RNAs (lnc-TALC) induces TMZ resistance via microglial complement components C5/C5a [70,71] ⑮ Glioma-secreted “hypoxic” exosomal IL-6 and miR-155-3p stimulate autophagy via STAT3 signalling and enhance CD163 and IL-10 expression in macrophages [44] ⑯ Glioma-derived IL-33 enhances activation of AIF1 (Iba1)+ resident microglia and recruitment of CD163+ BMDM [34]; microglial cells in the IL-33+ xenografts also expressed significant levels of IL-33 while upregulating pro-inflammatory cytokines, Spp1 and the lipid metabolism gene, *Apoe* followed by releasing a notable amount of the monocyte chemoattractant genes (*CCL2, CCL3,* and *CCL12*) ⑰ GSC-derived exosomes contain various immunosuppressive molecules that are part of the STAT3 pathway; GSCs-derived exosomes can induce CD163 expression in monocytes [72] Font colour: magenta, “Janus” factors that are secreted by both microglia/macrophages and GBM cells/GSCs (cf. Table 1). “Created with BioRender.com”.

**Figure 3 ijms-23-15612-f003:**
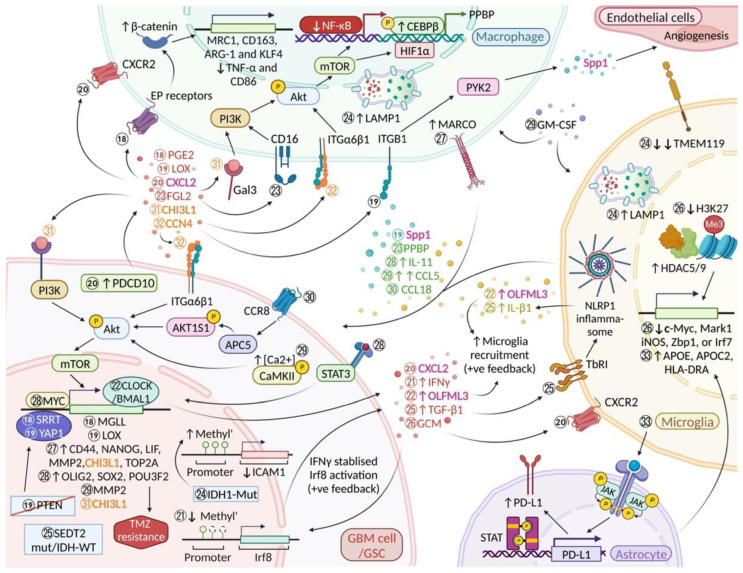
Key pathways involved in microglia/macrophage-glioma intercellular communication. Molecular event ⑱ Serrate RNA effector molecule homolog (SRRT) regulates stem cell-like properties in GSCs via transcription of the Monoacylglycerol lipase (*MGLL*) gene and stimulates a Prostaglandin E2 (PGE2)/E-prostanoid (EP) receptors/β-catenin mediated immunosuppressive phenotype in macrophages (expression of MRC1, CD163, ARG-1 and KLF4, and reduced TNF-α and CD86 expression) [73] ⑲ GBM cells lacking *PTEN* stimulate BMDM infiltration and activation via the Yes1 associated transcriptional regulator (YAP1)/ Lysyl oxidase (*LOX*)/ β1 Integrin (ITGB1)/Protein tyrosine kinase 2 (PYK2) pathway; BMDM-derived Spp1 sustains glioma cells survival and augments angiogenesis [36] ⑳ PDCD10-mediated recruitment of microglia and BMDM via CXCL2-CXCR2 signalling [37] ㉑ Mesenchymal GSCs mimic activation of Interferon regulatory factor 8 (*Irf8*) by erasing *Irf8* promoter methylation; glioma-derived IFNγ sustained activation of *Irf8* [74] ㉒ CLOCK-BMAL1 is involved in the recruitment of microglia (via OLFML3) and GSC renewal; microglia also express OLFML3 which may stimulate the recruitment of more microglial cells (feed-forward loop) [39,40] ㉓ Glioma-released Fibrinogen-like protein 2 (FGL2) induces secretion of Pro-platelet basic protein (PPBP) by a subset of TAMs through the activation of FGL2/CD16/PI3K/Akt/HIF1α signalling which promotes the expression of stem-like properties in glioma cells [75] ㉔ Isocitrate dehydrogenase (*IDH*)-mut glioma cells downregulate Intercellular adhesion molecule 1 (*ICAM1*) resulting in an increased expression of Lysosomal associated membrane protein 1 (LAMP1) in tumour-associated macrophages/microglia; significant decrease of microglia marker Transmembrane protein 119 (TMEM119) in *IDH1*-mut glioma [76] ㉕ Microglia activate the NLR family pyrin domain containing 1 (NLRP1) inflammasome and exhibit a pro-inflammatory and proliferative phenotype in SET domain containing 2, histone lysine methyltransferase (*SETD2*) mut/*IDH*-WT GBM upon stimulation by glioma-derived TGF-β1 [77] ㉖ Histone deacetylases (HDAC)5/9 mediates pro-inflammatory gene silencing (via H3K27me3) in microglia following GCM (glioma-conditioned medium) exposure [78] ㉗ Macrophage receptor with collagenous structure (MARCO)^high^ macrophages upregulate mesenchymal state-associated genes in GSCs [79] ㉘ Tumour-associated microglia/macrophage-derived IL-11 activates the STAT3-MYC pathway in GBM cells which in turn activate stemness-associated genes leading to higher tumorigenicity and temozolomide (TMZ) resistance [80] ㉙ TAM-derived CCL5 stimulates MMP2 production via the Calcium/calmodulin-dependent protein kinase II (p-CaMKII)-Akt pathway; glioma cells exhibit a strong affinity for Granulocyte-macrophage colony-stimulating factor (GM-CSF) activated GAMs [81] ㉚ Microglia/macrophages-derived CCL18 promotes glioma cell growth and invasion via the CCR8/Acid phosphatase 5 (ACP5)/AKT1 substrate 1 (AKT1S1)/Akt pathway [82] ㉛ CHI3L1-mediated intrinsic glioma cell signalling (CHI3L1/PI3K/Akt/mTOR) in a positive feedback loop promotes the infiltration of “pro-tumour” macrophages (paracrine; via CHI3L1/Gal3/PI3K/Akt/mTOR axis) [33] ㉜ GSC-derived CCN4 induces self-renewal and proliferation of GSCs (autocrine) and survival of glioma-associated macrophages (paracrine) through activation of the CCN4/ITGα6β1/Akt pathway [32] ㉝ Microglia induce an anti-inflammatory phenotype in reactive astrocytes via the JAK/STAT pathway; in turn, tumour-associated astrocytes mediate reprogramming of gene transcription in microglia [83] Methyl’: Methylation. Font colours: green, molecules secreted by both microglia and BMDMs; magenta, “Janus” factors that are secreted by both microglia/macrophages and GBM cells/GSCs; orange, key molecules acting on “Janus” pathways in microglia/macrophages and GBM cells/GSCs (cf. Table 1). “Created with BioRender.com”.

**Table 1 ijms-23-15612-t001:** Synergistic (“Janus”) genes and pathways that act in both glioblastoma cells and microglia/macrophages supporting glioblastoma progression in a complementary fashion (please also see Figures).

“Janus” Genes and Pathways		Molecular Mechanisms
CCN4/Integrin α6β1/ AKT	Cellular Communication Network Factor 4 Synonym: Wnt-induced signaling protein 1 (WISP1)	CCN4 facilitates self-renewal and proliferation of GSCs by binding to Integrin α6β1 on GSCs and activating the Akt pathway in an autocrine manner; CCN4 also supports the survival of glioma-associated macrophages via the CCN4/ITGα6β1/Akt pathway [32].
CHI3L1/Gal3/PI3K/AKT/mTOR → immunosuppressive phenotype in glioma associated-MΦ CHI3L1/PI3K/AKT → mTOR → glioma cell intrinsic signalling	Chitinase 3 Like 1	Galectin 3 (Gal3) binds to glioma-secreted CHI3L1 and promotes the infiltration of “pro-tumour” MΦ and appears to be regulated transcriptionally by NF-κB/CEBPβ in the CHI3L1/Gal3/PI3K/AKT/mTOR axis, which drives macrophages towards an immune suppressive phenotype [33]; CHI3L1 is also expressed in GSCs and regulated by the PI3K/Akt/mTOR pathway in a positive feedback loop [33].
SPP1	Secreted phosphoprotein 1 Synonym: Osteopontin (OPN)	GBM and glioblastoma stem cells (GSCs) use Spp1 to recruit macrophages into the TME [19]; a subset of microglia in the IL-33+ xenografts also expressed a significant amount of IL-33 and showed enrichment in pro-inflammatory cytokines, Spp1 [34]; Spp1 is highly expressed in GBM-infiltrating CD14+ BMDM [35]; the infiltrated BMDMs secrete Spp1 leading to extended glioma cells survival via inhibition of apoptosis [36]; infiltrated BMDMs secrete Spp1 and augment angiogenesis [36].
SELP-PSGL1	P-selectin P-selectin Glycoprotein ligand-1	Glioma-derived SELP reduces the phagocytic activity of microglia/BMDM, decreases their expression of inducible nitric oxide synthase (iNOS) and release of nitric oxide (NO) while increasing expression of IL-10 and TGF-β [22]; following exposure to soluble SELP (sSELP), a positive feedback loop causes overexpression of SELP and PSGL-1 by GBM and microglia cells [22].
CXCL2	Chemokine (C-X-C motif) ligand 2	Programmed cell death protein 10 (PDCD10) upregulation in GBM cells is followed by an increase in CXCL2 and resulting activation of CXCR2 in microglia [37]; CXCL2, which attracts granulocytes in mice [25], is highly unregulated by both glioma cells and microglia/macrophages; CXCL2, IL-8 and ANPEP (CD13) are considered alternative proangiogenic factors [27]; CXCL2 can promote angiogenesis independent of VEGF [38].
OLFML3	Olfactomedin Like 3	Clock circadian regulator (CLOCK), which is amplified in about 5% of GBMs, and Basic helix-loop-helix ARNT like 1 (BMAL1) are involved in the recruitment of microglia (via OLFML3) and GSC renewal [39]; Microglia also express OLFML3 [40] and can stimulate the recruitment of more microglial cells (feed-forward loop) [39,41]; Toedebusch et al. have proposed that microglia derived OLFML3 acts as a paracrine factor that facilitates glioma invasion [42]
IL-6	Interleukin-6	MSR1+ microglia/macrophages in glioblastoma show a pro-inflammatory phenotype and high IL-6 expression which is correlated with poor survival [18]; nuclear IL-33 facilitates tumour growth by triggering glioma-mediated expression of inflammatory cytokines (LIF, IL-6, IL-8, IL-1RN, IL-1β and secreted IL-33) [34]; GBM-associated endothelial cells are another important source of IL-6 in the TME [43]; IL-6 and miR-155-3p delivered by “hypoxic” exosomes derived from human GBM cells drive macrophages towards an immunosuppressive phenotype which supports glioma proliferation and migration [44].
miR-504	MicroRNA-504	When miR-504 is overexpressed in GSCs it not only inhibits their tumorigenic potential but modulates the stemness and mesenchymal transition of glioma stem cells. It further modulates their interaction with microglia where it is delivered via extracellular vesicles and promotes M1 polarization [45].
STIP1	Stress Induced Phosphoprotein 1	STIP1 is expressed by both glioma cells and microglia/macrophages and its increased expression correlates with glioma progression [46].
IL-33	Interleukin 33	De Boeck and colleagues [34] recently suggested that both the nuclear and secreted form of IL-33 are present within tumour cells in ~50% of human glioma specimens and GBM murine models; a subset of microglia in the IL-33+ xenografts also expressed a significant amount of IL-33 [34]. This makes IL-33 a “Janus” factor for many but not all gliomas.

## Data Availability

Not applicable.

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
