# Peer review of "Microglia and Brain Macrophages as Drivers of Glioma Progression"

_ijms, 2022, doi:10.3390/ijms232415612_

Round 1

Reviewer 1 Report

Zheng and Graeber review the roles of brain myeloid cells in glioblastoma, as well as their potential as treatment options. The review is well written and very thorough, covering and important topic. I have only very minor comments and recommend the publication of this manuscript in the International Journal of Molecular Sciences. My comments to clarify the manuscript in a few places are below.

1) line 193 - define WHO grade

2) paragraph 5.2 is very brief and the role (or expected role) of STIP1 could be expanded on slightly. The sentence that spans lines 334 and 335 should also be edited.

3) In section 14 - could the authors suggest some good reviews on aspects of therapy resistance they do not cover?

4) the sentence ending on line 884 should also be edited for clarity

Reviewer 2 Report

The article is obviously interesting, a lot of information and illustrations.

However, some points should be changed before the next step.

1. The abstract is too broad and does not have a structure.

2. In the manuscript the aim, methodology  are missed.

3. The conclusion should provide a clear opinion which is based on the review aim and not contain references.

Reviewer 3 Report

The review article by Zheng and Graeber focuses on understanding the mechanisms that allow malignant glioma cells to weaken microglia and brain macrophage defence mechanisms and surface molecules and cytokines that have a prominent role in microglia/macrophage-glioma cell interactions. The manuscript is well written and could benefit from addressing some of the comments below:

Because there are too many abbreviations, in order to make readers understandable, a list can be made.

The quality of Figure 2-3 is poor and could be improved.

Macrophage inflammatory proteins 1 alpha and 2 (MIP-1 alpha, MIP-2) are members of a growing family of cytokines thought to play a role in host defense. MIP-1 alpha and MIP-2 were previously shown to stimulate inflammatory cell recruitment. The authors may review this topic more in detail.

According to the activation state and functions of macrophages, they can be divided into M1 and M2 types. In order to make readers understandable, the authors may discuss this issue more in detail.

Line 40: Many roles of the class A macrophage scavenger receptor (SR-A) have been reported. SR-A displays the ability to bind and endocytose large quantities of modified lipoprotein. Hence, it is thought to be one of the main receptors involved in mediating lipid influx into macrophages, which promotes their conversion into foam cells that are abundant in the atherosclerotic lesion. A paragraph is required to describe its role and how it is related to this review article. 

Lines 53-54: Thus, MSR1+ microglia/BDMDs may support stem-like GBM cells and the progression of GBM in perivascular and necrotic niches. I suggest a paragraph to further explain the mechanisms underlying how MSR1+ microglia/BDMDs support stem-like GBM cells and the progression of GBM in perivascular and necrotic niches. More references should be cited to support this issue.

Line 314: Yu-Ju Wu et al. should read Wu.

Round 2

Reviewer 2 Report

No comments.

Reviewer 3 Report

The revised manuscript can be accepted for publication